# Autoantibody against Tumor-Associated Antigens as Diagnostic Biomarkers in Hispanic Patients with Hepatocellular Carcinoma

**DOI:** 10.3390/cells11203227

**Published:** 2022-10-14

**Authors:** Yangcheng Ma, Cuipeng Qiu, Bofei Wang, Xiaojun Zhang, Xiao Wang, Renato J. Aguilera, Jian-Ying Zhang

**Affiliations:** 1Department of Biological Sciences & NIH-Sponsored Border Biomedical Research Center, The University of Texas at El Paso, El Paso, TX 79968, USA; 2Department of Leukemia, The University of Texas at MD Anderson Cancer Center, Houston, TX 77030, USA

**Keywords:** tumor-associated antigen (TAA), hepatocellular carcinoma (HCC), Hispanic, autoantibody, immunodiagnostic biomarkers, serological proteome analysis (SERPA), driver gene

## Abstract

Background: Tumor-associated antigens (TAAs) have been investigated for many years as potential early diagnosis tools, especially for hepatocellular carcinoma (HCC). Nonetheless, very few studies have focused on the Hispanic HCC group that may be associated with distinct etiological risk factors. In the present study, we investigated novel anti-TAA autoantibodies as diagnostic biomarkers for Hispanic HCC patients. Methods: Novel TAA targets were identified by the serological proteome analysis (SERPA) and from differentially expressed HCC driver genes via bioinformatics. The autoantibody levels were validated by enzyme-linked immunosorbent assay (ELISA). Results: Among 19 potential TAA targets, 4 anti-TAA autoantibodies were investigated as potential diagnostic biomarkers with significantly high levels in Hispanic HCC sera, including DNA methyltransferase 3A (DNMT3A), p16, Hear shock protein 60 (Hsp60), and Heat shock protein A5 (HSPA5). The area under the ROC curve (AUC) value of the single autoantibodies varies from 0.7505 to 0.8885. After combining all 4 autoantibodies, the sensitivity of the autoantibody panel increased to 75% compared to the single one with the highest value of 45.8%. In a separate analysis of the Asian cohort, autoantibodies against HSPA5 and p16 showed significantly elevated levels in HCC compared to normal healthy controls, but not for DNMT3A or HSP60. Conclusion: Anti-DNMT3A, p16, HSPA5, and HSP60 autoantibodies have the potential to be diagnostic biomarkers for Hispanic HCC patients, of which DNMT3A and HSP60 might be exclusive for Hispanic HCC diagnosis.

## 1. Introduction

Liver cancer is the seventh most prevalent and second most deadly cancer worldwide [1]. It is estimated that by 2025, more than one million individuals will be affected by liver cancer annually [2]. The most common form of liver cancer is hepatocellular carcinoma (HCC) which compromises more than 80% of overall liver cancer cases [3]. Early diagnosis has been the key to the ideal prognosis of HCC for a long time, including serological biomarker screening. To date, the universal blood biomarker for HCC is the alpha-fetal protein (AFP) [4,5]. However, researchers have reported that the efficacy of AFP as an HCC diagnostic biomarker is not ideal due to the lack of specificity [6]. Numerous studies have demonstrated that the immune system can recognize tumor-associated antigens (TAA) from antigenic changes in tumor cells and produce autoantibodies as efficient biomarkers in multiple cancers [7,8]. It has been shown that the level of anti-TAA autoantibody could be drastically increased before the diagnosis point, affirming its value in the early diagnosis of HCC [9].

The incidence and mortality trends of HCC vary geographically and ethnically. As in the United States, the age-adjusted incidence of HCC in Hispanics has surpassed those of other ethnicities [10]. Metabolic disorders such as non-alcoholic fatty liver disease (NAFLD) and type 2 diabetes might be the leading cause of Hispanic HCC compared to other populations [11,12,13]. Although there is an urgent need to establish a surveillance system for the early diagnosis of Hispanic HCC patients, very few studies have focused on biomarkers exclusive to them. 

In the present study, we have investigated novel serological anti-TAA autoantibodies as biomarkers for the early diagnosis of Hispanic HCC. To achieve this goal, we investigated potential TAA targets using serological proteome analysis (SERPA) based on two-dimensional electrophoresis (2DE) from the whole cell lysate of HepG2 and SNU449 cell lines combined with mass spectrometry analysis. We also used bioinformatic tools to screen for HCC driver genes with significant mutations and differential mRNA expression as potential targets. Verification of the diagnostic value of the anti-TAA autoantibodies was accomplished by the enzyme-linked immunosorbent assay (ELISA) to determine whether the anti-TAA autoantibodies can differentiate liver cancer patients from normal healthy controls and patients with other liver diseases. In addition, the biomarkers were verified in the Asian sera cohort to see if the biomarkers are exclusive to Hispanic HCC patients. 

## 2. Materials and Methods

### 2.1. Serum Samples

The serum samples were obtained from the serum bank of the Cancer Autoimmunity Research Laboratory at the University of Texas at El Paso. The Hispanic sera cohort includes 24 samples with hepatocellular carcinoma (HCC), 20 samples with liver cirrhosis (LC), 26 samples with chronic hepatitis (CH), and 40 normal healthy sera (NHS) samples. Asian sera cohort includes 53 HCC samples, 20 LC samples, and 44 CH samples. This study was approved by the Institutional Review Board of UTEP.

### 2.2. Cell Culture

The HCC cell lines SNU449 (poorly differentiated cells) and HepG2 (well-differentiated cells) were purchased from American Type Culture Collection (ATCC, Manassas, VA, USA), and cultured in RPMI 1640 (HyClone, Logan, UT, USA) and DMEM (Dulbecco’s modified Eagle’s medium) (HyClone, Logan, UT, USA), respectively, which were supplemented with 10% fetal bovine serum (FBS), 100 units/mL penicillin and 100 unite/mL streptomycin. Cells were grown in 75 cm^2^ tissue culture flasks (Corning, Glendale, AZ, USA) and allowed to reach 95% confluency. When collecting cells, cells were rinsed with DPBS (Gibco, Carlsbad, CA, USA) 3 times and then incubated with trypsin-EDTA solution (Gibco, Carlsbad, CA, USA) and harvested in a 15 mL centrifuge tube for further study.

### 2.3. Two-dimensional Gel Electrophoresis 

HepG2 and SNU449 cell lysates were prepared and lysed in the rehydration sample buffer (Bio-Rad, Hercules, CA, USA). After being vigorously vortexed for 90 min at room temperature (RT), the samples were next centrifuged at 16,000× *g* under 4 °C temperature for 20 min. The supernatant was collected, and the protein concentration was measured by Bradford Protein Assay kit (Bio-Rad, Hercules, CA, USA). In the first-dimensional (1DE) analysis, 125 μL of 150 μg protein was loaded onto the isoelectric focusing (IEF) strip (Non-linear, pH 3–10, 7 cm, Bio-Rad, Hercules, CA, USA). Then, 1DE was performed in the PROTEAN IEF (Bio-Rad, Hercules, CA, USA) with linearly increased voltage from 0 to 250 V in 30 min, then gradually increased voltage to 400 V in 90 min, and finally kept at 4000 V for 25,000 kvh. The prepared IEF strips were subsequently used for the second-dimensional (2DE) gel electrophoresis. Strips were docked into 12% SDS-polyacrylamide (SDS-PAGE) gel by Overlay agarose (Bio-Rad, Hercules, CA, USA), and the electrophoresis was accomplished under a constant voltage of 80 V for 2 h. The SDS-PAGE gels were then transferred onto the nitrocellulose (NC) membrane (Osmonics Inc., Westborough, MA, USA) or stained with 0.1% Coomassie blue R-250 reagent. The protein spots were identified by PDQuest 2-DE analysis software v. 8.0(Bio-Rad, Hercules, CA, USA).

### 2.4. Western Blotting Analysis

After the 2DE was conducted, the SDS-PAGE gel was transferred onto the NC membrane under a constant current of 250 mA for 70 min. Non-fat milk (Bio-Rad, Hercules, CA, USA) was diluted in 0.1% tris-buffered saline tween (TBST) to 5% to be used as the blocking reagent. The NC membranes were blocked overnight at 4 °C. The positive sera pool or the negative sera pool was diluted with 5% non-fat milk at a concentration of 1:200 and incubated with the NC membranes for 2 h under RT. Horse-radish peroxidase (HRP)-conjugated goat anti-human IgG (Caltag Laboratories, San Francisco, CA, USA) was diluted with 5% non-fat milk at a concentration of 1: 10,000 and was subsequently incubated with the NC membrane for 1 h at RT. The blotting spots were detected by the enhanced chemiluminescence (ECL) kit (Amersham, Arlington Heights, IL, USA) with iBright (ThermoFisher, Waltham, MA, USA).

### 2.5. Mass Spectrometry Analysis

The mass spectrometry analysis was executed by Applied Biomics (Hayward, CA, USA). The modified porcine trypsin protease (Trypsin Gold, Promega, Madison, WI, USA) was used to digest target protein spots. The digested tryptic peptides were desalted by Zip-tip C18 (Millipore, Burlington, MA, USA) and were eluted from the Zip-tip with 0.5 uL of matrix solution (a-cyano-4-hydroxycinnamic acid (5 mg/mL in 50% acetonitrile, 0.1% trifluoroacetic acid, 25 mM ammonium bicarbonate) and spotted on the AB SCIEX MALDI plate (Opti-TOFTM 384 Well Insert). MALDI-TOF MS and TOF/TOF tandem MS/MS were performed on an AB SCIEX TOF/TOF™ 5800 System (AB SCIEX, Framingham, MA, USA). An average of 4000 laser shots per spectrum was used for MALDI-TOF. TOF/TOF tandem MS fragmentation spectra were acquired for each sample, averaging 4000 laser shots per fragmentation spectrum on each of the 10 most abundant ions present in each sample (excluding trypsin autolytic peptides and other known background ions). Both of the resulting peptide mass and the associated fragmentation spectra were submitted to the GPS Explorer workstation equipped with MASCOT search engine v. 2.0 (Matrix Science, Boston, MA, USA) to search the Swiss-Prot database. Candidates with either protein score C.I.% or Ion C.I.% greater than 95 were considered significant.

### 2.6. Enzyme-linked Immunosorbent Assay (ELISA)

Recombinant proteins were commercially purchased: Merlin, SETDB1, BRG1, RNA helicase A, CALR, and TPM3 were purchased from Signalway Antibody (Greenbelt, MD, USA); HSPA5, ACTG1, ENO1, NRAS, ERK2, p16, and GMPS were purchased from Lifespan Biosciences (Seattle, MA, USA); HSP60, HSP70, and TPI were purchased from Abcam (Boston, MA, USA); GNAS and P4HB were purchased from Aviva Systems Biology (San Diego, CA, USA); DNMT3A was purchased from Novus Biologicals (Centennial, CO, USA). The proteins’ purity was >90%. When coating onto the polystyrene 96-well plates, the proteins were diluted to a final concentration of 0.75 μg/mL in pH 7.2 phosphate-buffered saline (PBS) (Gibco, Waltham, MA, USA), each well was loaded with 100 µL of diluted protein. The 1:200 diluted serum samples were used as primary antibodies for 100 μL/well and stayed at 37 °C for 1.5 h. Next, the plates were rinsed with PBS 3 times, 1:1000 HRP-conjugated goat anti-human IgG was used as the secondary antibody for 100 µL/well and stayed at 37 °C for 1 h. After another 3 times of PBS washes, the chromogenic substrate ABTS (2,2′-azinobis (3-ethylbenzthiazoline-6-sulfonic acid)) (Alfa Aesar, Tewksbury, MA, USA) was applied and an average optical density (OD) of 405 nm was used for plate reading.

### 2.7. Bioinformatic Analysis 

Driver genes of HCC were acquired from the IntoGen database and were subsequently screened by the Oncomine database (http://www.oncomine.org/, accessed on 9 August 2021). The genes in HCC groups retaining an mRNA expression greater than 1.5-fold change and p value less than 0.001 in no less than 2 cohorts were deemed as hits. The hits were validated in a separate The Cancer Genome Atlas (TCGA) database via the the University of Alabama at Birmingham Cancer data analysis Portal (UALCAN) algorithm (http://ualcan.path.uab.edu/, accessed on 9 August 2021). The hits with significantly higher transcriptional expression in the HCC group than in the normal group with a *p* value less than 0.001 will be deemed as potential TAA targets. 

### 2.8. Statistical Analysis

The cutoff value in the results of ELISA tests was defined as the mean of optical density (OD) value plus 2 times of standard deviation of OD value from the normal healthy group. Non-parametrical Mann–Whitney test was used to compare the mean OD value between 2 different groups. The diagnostic value of anti-TAA autoantibodies was represented by receiver operating characteristic (ROC) analysis of single variables, leading to the estimates of area under ROC curve (AUC) value with a 95% confidence interval (CI). Positive frequencies of each group were compared by chi-square test. Statistical analysis was accomplished by SPSS v.27.0.1 (IBM, NY, USA) and GraphPad Prism v. 9.4.1 software (GraphPad, CA, USA). Differences were considered statistically significant, with a significant level (*p*) less than 0.05.

## 3. Results

### 3.1. Nine Potential TAA Targets were Identified by Serological Proteome Analysis (SERPA)

Autoantibodies against cellular antigens are commonly found in systemic autoimmune diseases and have been described as effective diagnostic and prognostic parameters in cancers. To narrow the sample number and to elevate the sensitivity of the immunoproteomic tests, we performed Western-Blotting analysis and indirect immunofluorescence assay to screen out “positive sera” with relatively high autoantibody expression from Hispanic HCC. From 24 Hispanic HCC serum samples and 40 Hispanic normal healthy sera (NHS) samples, 8 positive sera and 8 negative sera were selected (Appendix A). Both sera groups were equally mixed and formed as “positive” and “negative” sera pool for subsequent use. 

The SERPA with two-dimensional electrophoresis (2DE) was applied to screen potential TAA targets. Our group previously used the immunoproteomic approach and has successfully identified TAAs as biomarkers in various cancers [14,15]. Total protein obtained from HepG2 and SNU449 cell lysates were separated on the SDS-PAGE gel by their isoelectric points and molecular weights and then either stained with Coomassie blue or transferred onto the nitrocellulose (NC) membrane. After performing the experiments in triplicate, protein spots that showed significantly stronger immunoblot stains on the NC membranes that were incubated with HCC positive sera group were compared to the negative sera group. Differentially expressed proteins were traced back to the corresponding SDS-PAGE gels (Appendix A). The gel spots were then excised for further MALDI-TOF/TOF tandem mass spectrometry analysis. Of the 24 analyzed samples, 21 of them had both matched molecular weight and isoelectric point (pI). After eliminating the replicates, there were nine potential TAA targets identified by 2DE, which were HSPA5, HSP60, TPI, P4HB, ACTG1, HSP70, ENO1, TPM3, and CALR (Table 1.).

### 3.2. Ten Potential TAA Targets Were Identified from HCC Driver Genes

Driver genes refer to the genes with great mutation tendency in specific cancers, predicting that they could drive the tumorigenesis progression. A previous study systematically discovered driver genes in multiple cancers and developed an integrated cancer driver gene database named Integrative OncoGenomics (IntOGen) [16]. In hepatic cancer, 1616 samples from 9 database cohorts were collected, and 75 driver genes were identified from the pipeline. Bioinformatic algorithms were used to investigate the most differentially expressed driver genes as proper potential TAA targets. Within eight cohorts from the Oncomine database, if the mRNA expressions of the driver gene between the HCC tissue groups and the normal liver tissue groups showed a fold change of bigger than 1.5, and the cohorts’ number no less than 2, then the driver gene was deemed as a hit. The hits were then verified in another The Cancer Genome Atlas (TCGA) cohort through the the University of Alabama at Birmingham Cancer data analysis Portal (UALCAN) database. If the mRNA expression between the HCC and normal liver groups was significantly different (*p* < 0.001), then the hits will be confirmed as potential TAA targets. Eventually, 10 potential TAA targets were investigated from 75 HCC driver genes, including p16, SETDB1, RNA helicase A, BRG1, GNAS, Merlin, DNMT3A, NRAS, GMPS, and ERK2 (Table 2.).

### 3.3. Frequencies and Titers of Autoantibodies against 19 Potential TAA Targets by ELISA

Enzyme-linked immunosorbent assay (ELISA) was applied to investigate the autoantibody against potential TAA targets that were investigated by the immunoproteomic approach and driver genes. The recombinant protein products of the potential TAA targets were used as coating antigens, and a cohort of Hispanic sera was used as testing samples, including 24 HCC, 20 liver cirrhosis (LC), 26 chronic hepatitis (CH), and 40 normal healthy sera (NHS) samples. The cutoff value was established as the mean plus 2 times standard deviations (SD) of the NHS group’s optical density (OD) value. To investigate the frequency of autoantibodies against all the TAA targets, the positive rates above the cutoff values of the autoantibodies are listed in Table 3. Among the 19 potential TAA targets, 4 TAAs showed significantly higher autoantibody frequencies in the HCC group compared to the NHS group, which are DNMT3A, p16, HSP60, and HSPA5. We next compared the autoantibody titer distribution between the HCC group and 3 other groups within the 4 TAAs. The autoantibody titers in the HCC group are significantly higher than the NHS group in all 4 TAAs, while in HSP60 and p16 they were also higher than both the CH group and the NHS group (Figure 1). It was also determined that in DNMT3A and HSP60 the autoantibody titers of the LC group are higher than the HCC group. The results indicate that these 4 anti-TAA autoantibodies are able to discriminate Hispanic HCC from the normal healthy group and can be used as potential diagnostic biomarkers for Hispanic HCC patients. 

### 3.4. Diagnostic Value of Anti-TAA Autoantibodies and Autoantibody Panel as Biomarkers for Hispanic HCC 

As shown in Figure 2, the receiver operating characteristic (ROC) analysis showed that the anti-TAA autoantibody could differentiate the Hispanic HCC group from normal Hispanic controls, with the area under the ROC curve (AUC) value varying from 0.7505 to 0.8885 and *p* values less than 0.001. P16 and HSP60 showed significant diagnostic models with AUC values of 0.6875 and 0.6862, respectively, in comparing HCC and CH (*p* < 0.05). In HCC vs. LC, DNMT3A and HSP60 showed significant differentiation with AUC values of 0.6979 and 0.748. We next evaluated a panel of four anti-TAA autoantibodies as a diagnostic platform for Hispanic HCC. The autoantibodies were sequentially added to the panel, from DNMT3A, which has the highest frequency of antibody, to HSPA5 with the lowest frequency (Table 4). After adding all four autoantibodies, the panel’s sensitivity increased to 75%, compared to the highest single one (45.8%). The specificity of the panel decreased from 95% to 85%. The anti-TAA autoantibody panel’s positive predictive value (PPV) and negative predictive value (NPV) were 83.3% and 77.3%, respectively. The likelihood ratio (LR) reflects the potentiality of a diagnostic platform and how likely a patient has a disease or condition. The higher the LR+, the more assurance that the person is a valid patient; the lower the LR-, the more confidence that the person does not retain the specific health problem. For the 4-TAA panel, the LR+ and LR- were 5.00 and 0.29, respectively, indicating that the positive result is likely to originate from patients with HCC, and the negative consequence is insufficient to rule out HCC. 

### 3.5. Validation of Four Anti-TAA Autoantibodies in a Separate Asian Sera Cohort

To validate the robustness of the four autoantibodies biomarkers, we used another cohort of Asian serum samples containing 53 HCC, 20 LC, and 44 CH, to verify if those autoantibodies are exclusive to Hispanic HCC patients. To maintain consistency, we used the same NHS sera control group and the same criteria for setting up cutoff value as in the previous section. The frequencies of the anti-TAA autoantibodies are shown in Table 5. Among all the Asian HCC sera groups, only anti-HSPA5 autoantibody showed a significantly higher positive rate (28.3%) in the HCC group than the NHS group (*p* = 0.003). Concerning the other groups, the anti-p16 and anti-HSPA5 autoantibody showed greater expression for the LC groups, while no significantly higher autoantibody frequency was detected in the CH groups. We next compared the OD values between the Asian HCC with the LC, CH, and NHS groups (Figure 3). The results showed that anti-p16 and HSPA5 autoantibodies showed significantly higher OD values in HCC compared to the NHS group. Although anti-DNMT3A autoantibody were found elevated in the LC compared to the HCC group, however, no significant differences were found between HCC and NHS groups in anti-DNMT3A or HSP60 autoantibodies, suggesting that these two autoantibodies might not be applicable for Asian HCC diagnosis, and they might be the exclusive diagnostic biomarkers for Hispanic HCC.

## 4. Discussion

Numerous studies have noted the importance of biomarkers in the early diagnosis of cancers. TAAs, being efficient and robust biomarkers, have been applied in HCC diagnosis for decades [17,18]. Nonetheless, there is no research focused on Hispanic HCC patients. In the United States, the highest incidence rate and lowest overall survival rate of HCC were observed in the Hispanic population compared to other ethnic groups [19,20]. According to previous studies, non-alcoholic fatty liver disease (NAFLD) and other metabolic risk factors such as obesity and diabetes might be the major etiological causes of Hispanic HCC, suggesting its special pathogenesis mechanisms [20,21]. Therefore, it is urgently required to find novel approaches for diagnosing and treating Hispanic HCC patients. This study is the first one to identify anti-TAA autoantibodies as biomarkers exclusive to the diagnosis of Hispanic HCC patients. To identify specific biomarkers, an immunoproteomic approach named SERPA was used in our study to discover differentially expressed antigens from Hispanic HCC sera compared to normal healthy controls. Such an approach has been successfully applied in our group to discover novel biomarkers within different malignancies [14,22]. In the 2DE part of the SERPA analysis, we used not one but two HCC cell lines, including HepG2 and SNU449 cells, compared to previous SERPA studies, which only used the HepG2 cell line. By combing the two cell lines, both well differentiated and poorly differentiated HCC cells were included, which should result in a more integrated landscape of HCC proteome [23,24]. Driver genes refer to those mutational genes that strongly correlate with tumorigenesis [25]. Previous studies have suggested that new antigenic epitopes of TAAs could be produced via somatic mutations, and can be detected by the immune system and bound by autoantibodies in both mutated and non-mutated forms [26,27]. To investigate the linkage between driver genes and Hispanic HCC TAAs, we are aiming to filter proper TAA candidates from the driver gene database IntOGen [16]. With the acknowledgment that genes highly expressed in HCC tissue compared to normal liver tissue are considered as proper TAA candidates, we verified all 75 driver genes from the IntOGen database via the Oncomine and UALCAN databases and found another 10 candidates, which provided a more integrated screen for novel TAA discovery [28,29]. Among all 19 potential TAA targets, 4 were found meaningful with significantly higher positive rates of autoantibodies in the HCC group compared to the NHS group, including DNMT3A, p16, HSPA5, and HSP60.

Of the four TAAs that we identified, DNMT3A retains the highest sensitivity. As a family member of DNA methyltransferases, DNMT3A can catalyze de novo DNA methylation in CpG islands of gene promoters, resulting in the silencing of the tumor suppressor genes in multiple cancers [30]. Hassona et al. have indicated an elevated mRNA expression of DNMT3A in HCC patients with an AUC of 0.958 in discriminating HCC from hepatitis cirrhotic patients [31]. Other studies have shown that DNMT3A could be involved in the HCC tumorigenesis in multifaceted ways, including the PTEN/Akt pathway and microRNA and lncRNA expressions [32,33]. In addition, DNMT3A-mediated epigenetic silencing of PTEN was found in NAFLD-induced HCC, suggesting the linkage between DNMT3A expression and the specific etiology of Hispanic HCC [34]. P16, the product of the CDKN2A gene, is another TAA biomarker we identified from the Hispanic HCC group. Studies have investigated that the methylation of the CDKN2A gene and the dysregulation of p16 expression could lead to HCC progression [35,36]. It was believed that homozygous deletion of the CDKN2A gene would result in the inactivation of the G1 to S phase cell cycle and thus contribute to HCC tumorigenesis [37]. The capability of p16 as an HCC biomarker has been reported in the previous literature, Zhang et al. and Koziol et al. reported p16 as a member of the TAA microarrays and it has great performance in the early diagnosis of HCC [18,38]. Wang et al. also reported an AUC value of 0.62 for circulating anti-p16 antibodies in the plasma of HCC patients [39]. The heat shock protein (HSP) family has also been shown to correlate with multiple malignancies. Known to function as molecular chaperones, HSP60 and HSPA5 assist the folding of newly synthesized and misfolded proteins [40]. Our lab has identified HSP60 and HSPA5 as valid TAA biomarkers in the diagnosis of HCC before [14,41]. Previous research also verified the efficacy of those two proteins as HCC diagnostic biomarkers [42,43]. Interestingly, hepatocyte ER stress, which is regulated by HSPA5, has been suggested to play an important role in the development of steatohepatitic HCC [44], which accounts for a great proportion of the Hispanic HCC. 

By comparing the OD values of each autoantibody between HCC and NHS groups, all four autoantibodies showed significant differences, which validated that those anti-TAA autoantibodies could be potential serological diagnostic biomarkers for Hispanic HCC. P16 and HSP60 can also differentiate the HCC group from the CH group. As the primary progression from normal liver to liver cancer, CH patients involve minor immune alteration and structural modification compared to healthy individuals. By showing higher titers in HCC compared to CH and NHS, anti-p16 and HSP60 autoantibodies can better discriminate cancer patients against the early progression of the disease, thus providing a more accurate diagnostic prediction. Results showed that the autoantibodies against DNMT3A and HSP60 showed higher expression in the LC group compared to the HCC, one possible reason could be that most of the LC patients retained the pre-cancer status and were prone to generate greater anti-TAA immune reactivities. Prior studies have shown that autoantibody expression will experience a sudden flux before the diagnosis of liver cancer and predicts the diagnosis of liver cancer with a lead time of 0.75 years [9,38]. Therefore, we are confident that the four autoantibodies could be potential early diagnostic biomarkers for Hispanic HCC patients. The ROC results showed consistent trends with the autoantibodies titers. All four autoantibodies showed AUC values higher than 0.75 in the HCC vs. NHS models with p values less than 0.001, indicating the significant efficacies of the diagnostic models. By combing all four anti-TAA autoantibodies, the sensitivity of the autoantibody panel reached 75%, which was drastically elevated compared to the single TAA with the highest value. Although the model’s specificity decreased from 95% to 85%, the autoantibody panel still showed a greater diagnostic value, suggesting its application in future clinical applications. In the present study, anti-p16 and HSPA5 autoantibodies were found elevated in Asian HCC than in NHS, suggesting that both biomarkers might be universal to the HCC regardless of the ethnic differences. In contrast, the anti-DNMT3A and HSP60 autoantibodies could be exclusive to Hispanic HCC patients. This can partially attribute to the Hispanic HCC proteome alteration and the specific mechanism of Hispanic HCC tumorigenesis.

There remain some limitations within the present study. First, the serum sample number could be higher. In the current study, we have 110 samples in the Hispanic sera cohort and 117 samples in the Asian sera cohort, which is limited for research regarding ethnic populations. To further test the validity of the TAAs, a larger cohort of Hispanic sera is needed, which could reduce the bias and include a more integrated cancer proteome. Another weakness of the current research is that it lacks the verification of the four TAAs in ethnic populations other than Asian, for instance, Caucasian white and African American. Like the Hispanic group, the African American harbors a high prevalence of metabolic disorders and a high incidence of HCC [20]. By comparing the Hispanics to other populations, the TAA biomarker exclusive to the Hispanic HCC will be investigated, and a complete biomarker disparity will be shown for future studies. 

In conclusion, we identified 4 out of 19 anti-TAA autoantibodies as biomarkers in Hispanic HCC, including DNMT3A, p16, HSP60, and HSPA5. The panel of the four anti-TAA autoantibodies showed greater diagnostic value than the single ones. By comparing with the Asian sera cohort, anti-DNMT3A and HSP60 autoantibodies might be exclusive for diagnosing Hispanic HCC patients. 

## Figures and Tables

**Figure 1 cells-11-03227-f001:**
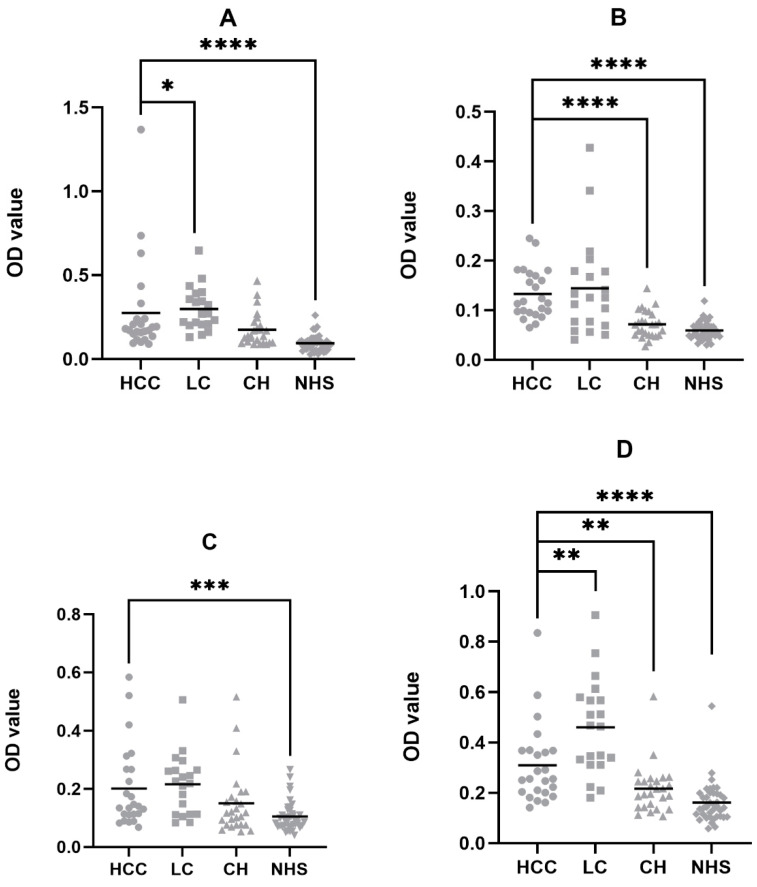
Autoantibody titer distributions of the 4 TAAs in the Hispanic HCC, LC, CH, and NHS groups by ELISA. Mann–Whitney tests show significant optical density (OD) differences between HCC and NHS groups in all 4 anti-TAA autoantibodies. Significant differences were identified between HCC against CH in p16 and HSP60, and HCC against LC in DNMT3A and HSP60. (**A**) DNMT3A; (**B**) p16; (**C**) HSPA5; (**D**) HSP60. HCC: Hepatocellular carcinoma; LC: Liver Cirrhosis; CH: Chronic Hepatitis; NHS: Normal Healthy Sera. * *p* < 0.05; ** *p* < 0.01; *** *p* < 0.001; **** *p* < 0.0001.

**Figure 2 cells-11-03227-f002:**
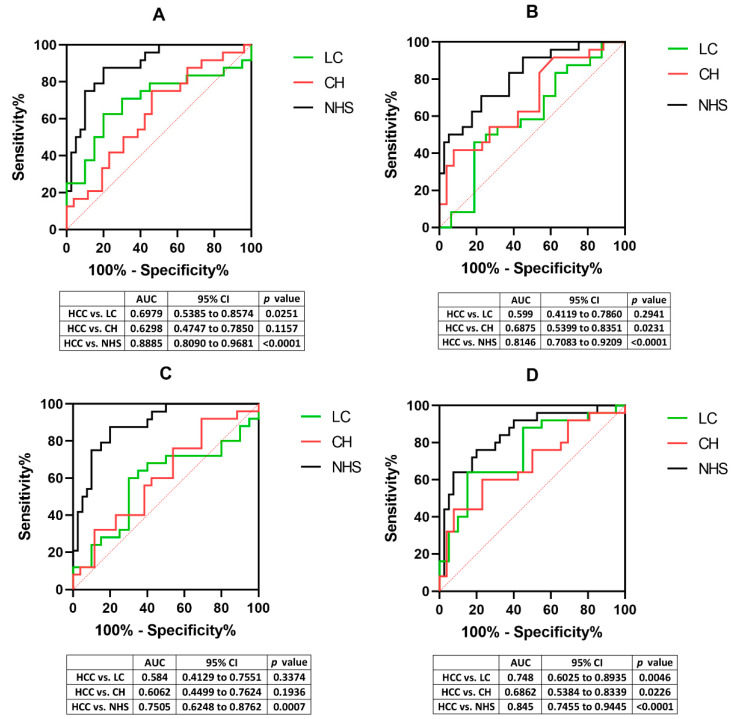
ROC analysis of anti-TAA autoantibodies between Hispanic HCC and LC, CH, and NHS groups. *p* value less than 0.05 indicates the confidence of the diagnostic model. (**A**) DNMT3A; (**B**) p16; (**C**) HSPA5; (**D**) HSP60. HCC: Hepatocellular carcinoma; LC: Liver Cirrhosis; CH: Chronic Hepatitis; NHS: Normal Healthy Sera. AUC: area under the receiver operating characteristic (ROC) curve.

**Figure 3 cells-11-03227-f003:**
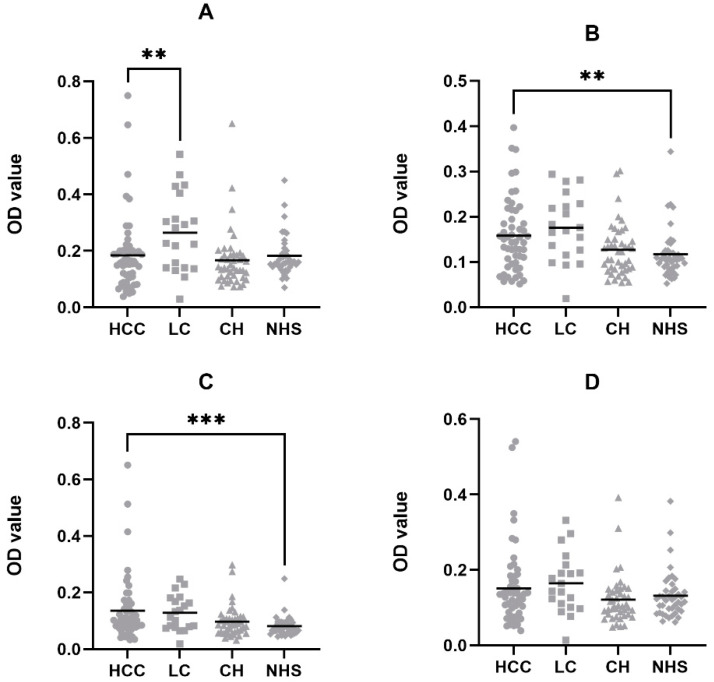
Autoantibody titers of the four TAAs in the Asian sera cohort by ELISA. The autoantibody titers were compared between the Asian HCC group with the LC, CH, and NHS sera groups. Mann–Whitney tests show significant differences in anti-p16 and HSPA5 autoantibodies when comparing the Asian HCC group with the NHS control group. (**A**) DNMT3A; (**B**) p16; (**C**) HSPA5; (**D**) HSP60. HCC: Hepatocellular carcinoma; LC: Liver Cirrhosis; CH: Chronic Hepatitis; NHS: Normal Healthy Sera. ** *p* < 0.01, *** *p* < 0.001.

**Table 1 cells-11-03227-t001:** Nine potential TAA targets identified by SERPA.

Identified Protein Name/ Protein Symbol	Molecular Weight (kDa)	Isoelectric Point	Origination	Function
Endoplasmic reticulum chaperone BiP/HSPA5	78	5.07	HepG2/SNU449	Protein folding and misfolded proteins degradation
60 kDa heat shock protein/HSP60	60	5.70	HepG2/SNU449	Mitochondrial protein import and macromolecular assembly
Triosephosphate isomerase/TPI	26	6.45	HepG2/SNU449	Catalyzes metabolites interconversion in glycolysis and gluconeogenesis
Protein disulfide-isomerase/P4HB	57	4.76	HepG2/SNU449	Disulfide bonds catalyzation
Actin, cytoplasmic 2/ACTG1	41	5.31	HepG2	Highly conserved protein in various cells
Heat shock 70 kDa protein/HSP70	70	5.11	SNU449	Molecular chaperone that surveillance protein quality control
Alpha-enolase/ENO1	47	7.01	HepG2	Glycolytic enzyme and growth controller
Tropomyosin alpha-3 chain/TPM3	32	4.68	HepG2	Regulates calcium-dependent muscle contraction
Calreticulin/CALR	48	4.29	HepG2	Calcium-binding chaperone that promotes protein folding

**Table 2 cells-11-03227-t002:** Ten potential TAA targets were identified from seventy-five driver genes.

Gene Name/ Protein Symbol	Mutation Counts	Mutated Samples	Mutation Types	Cohorts Number With Fold Change > 1.5	Function
CDKN2A/ p16	34	29	Missense (70%), truncating (26%), synonymous (14%)	2	Induce cell cycle arrest in G1 and G2 phases
SETDB1/ SETDB1	32	12	Missense (71%), synonymous (19%), truncating (10%)	3	Histone methyltransferase
DHX9/ RNA helicase A	33	11	Missense (84%), synonymous (13%), truncating (13%)	2	Nucleic acid helicase that unwinds DNA and RNA
SMARCA4/ BRG1	30	9	Missense (62%), synonymous (21%), truncating (17%)	3	Transcriptional activator
GNAS/GNAS	26	8	Missense (65%), synonymous (23%), truncating (8%), splice-site (4%)	2	Key component of adenylyl cyclase signal transduction pathways
NF2/ Merlin	14	6	Missense (50%), truncating (42%), synonymous (8%)	3	Regulator of Sav/Wts/Hpo signaling pathway
DNMT3A/ DNMT3A	25	4	Missense (52%), synonymous (22%), truncating (22%), splice-site (4%)	2	DNA methylation
NRAS/ NRAS	12	4	Missense (91%), synonymous (9%)	2	Bind GDP/GTP and possess intrinsic GTPase activity
GMPS/ GMPS	9	4	Missense (100%)	4	Catalyzes the conversion of XMP to GMP
MAPK1/ ERK2	10	3	Missense (90%), synonymous (10%)	2	Serine/threonine kinase in MAP signal transduction pathway

**Table 3 cells-11-03227-t003:** Frequencies of autoantibodies against nineteen potential targets.

Antigens	HCC (n = 24)	LC (n = 20)	CH (n = 26)	NHS (n = 40)
HSPA5	8 (33.3%) *	12 (60.0%) **	4 (15.4%)	3 (7.5%)
HSP60	9 (37.5%) **	15 (75.0%) **	2 (7.7%)	1 (2.5%)
TPI	2 (8.3%)	1 (5.0%)	0 (0.0%)	0 (0.0%)
P4HB	6 (25.0%)	10 (50.0%) **	2 (7.7%)	3 (7.5%)
ACTG1	4 (16.7%)	6 (30.0%) *	1 (3.8%)	2 (5.0%)
HSP70	1 (4.2%)	14 (70.0%) **	0 (0.0%)	2 (5.0%)
ENO1	4 (16.7%)	10 (50.0%) **	0 (0.0%)	2 (5.0%)
TPM3	5 (20.8%)	2 (10.0%)	1 (3.8%)	2 (5.0%)
CALR	3 (12.5%)	0 (0.0%)	0 (0.0%)	2 (5.0%)
P16	10 (41.7%) **	9 (45.0%) **	6 (23.1%) *	1 (2.5%)
SETDB1	3 (12.5%)	4 (20.0%)	1 (3.8%)	1 (2.5%)
RNA helicase A	4 (16.7%)	1 (5.0%)	2 (7.7%)	3 (7.5%)
BRG1	0 (0.00%)	2 (10.0%)	1 (3.8%)	1 (2.5%)
GNAS	4 (16.7%)	1 (5.0%)	2 (7.7%)	2 (5.0%)
NF2	2 (8.3%)	5 (25.0%)	0 (0.0%)	2 (5.0%)
DNMT3A	11 (45.8%) **	17 (85.0%) **	5 (19.2%)	2 (5.0%)
Merlin	5 (20.8%)	7 (35.0%) **	4 (15.4%)	2 (5.0%)
GMPS	1 (4.2%)	2 (10.0%)	1(3.8%)	1 (2.5%)
ERK2	3 (12.5%)	15 (75.0%) **	4 (15.4%)	2 (5.0%)

HCC: hepatocellular carcinoma; LC: liver cirrhosis; CH: chronic hepatitis; NHS: normal healthy sera. * *p* < 0.05, ** *p* < 0.01, cutoff value = (mean + 2 SD).

**Table 4 cells-11-03227-t004:** Stepwise addition of autoantibodies to the panel.

Antigens		HCC	NHS	Se	Sp	FP	FN	PPV	NPV	LR+	LR-
1 TAA	DNMT3A	45.8%	5%	45.8%	95%	5%	54.2%	90.2%	63.7%	9.16	0.57
2 TAA	DNMT3A or P16	58.3%	7.5%	58.3%	92.5%	7.5%	41.7%	88.6%	68.9%	7.77	0.45
3 TAA	DNMT3A or P16 or HSP60	70.8%	10%	70.8%	90%	10%	29.2%	87.6%	75.5%	7.08	0.32
4 TAA	DNMT3A or P16 or HSP60 or HSPA5	75%	15%	75%	85%	15%	25%	83.3%	77.3%	5.00	0.29

HCC: Hepatocellular carcinoma; NHS: Normal healthy sera; Se: sensitivity = positive/ number of HCC; Sp: specificity = negative/ number of NHS; FP: false positive = 1 − Sp; FN: false negative = 1 − Se; PPV: positive predicitive value = Se/(Se + FP); NPV: negative predictive value = Sp/(Sp + FN); LR+: positive likelihood ratio = Se/(1 − Sp); LR-: negative likelihood ratio = (1 − Se)/Sp.

**Table 5 cells-11-03227-t005:** Frequency of the four anti-TAA autoantibodies in Asian sera cohort.

Antigens	Asian HCC (n = 53)	Asian LC (n = 20)	Asian CH (n = 44)	NHS (n = 40)
DNMT3A	5 (9.4%)	5 (25%)	3 (6.8%)	2 (5%)
P16	9 (17.0%)	7 (35.0%) *	4 (9.1%)	3 (7.5%)
HSPA5	15 (28.3%) **	8 (40%) ***	4 (9.1%)	1 (2.5%)
HSP60	6 (11.3%)	3 (15%)	3 (6.8%)	2 (5%)

HCC: hepatocellular carcinoma; LC: liver cirrhosis; CH: chronic hepatitis; NHS: normal healthy sera. * *p* < 0.05, ** *p* < 0.01, *** *p* < 0.001, cutoff value = (mean + 2 SD).

## Data Availability

Data are available upon request.

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
