# Peer review of "Autoantibody against Tumor-Associated Antigens as Diagnostic Biomarkers in Hispanic Patients with Hepatocellular Carcinoma"

_cells, 2022, doi:10.3390/cells11203227_

Round 1
Reviewer 1 Report
The authors of the manuscript provided an excellent research study with valuable results to the scientific community. The methods the authors approached are clearly presented and described. The results they obtained are well compared with other results found in the literature. Overall, I recommend that the manuscript should be published, with minor check of English spelling/grammar and the references.
Reviewer 2 Report
Dear authors of the draft titled "Autoantibody against tumor-associated antigens as diagnostic biomarkers in Hispanic patients with hepatocellular carcinoma", your manuscript presented putative anti-TAA autoantibodies as diagnostic biomarkers for Hispanic HCC.
Yet, the data presented here are not sufficient to support the authors claims and therefore I suggest a major revision.
Minor Issue:
- Page 4 lane 149 it states 400x magnification, please verify.
- figure legends must be improved in technical details.
- paragraph "Ten potential TAA targets were identified from HCC driver genes", please showed the fold change of the selected genes, and stratify them per mutation type (missense, CNV....).
- Fig.3 is not readable, try to remove the numbers and try to highlight better the differences between the samples. Moreover, fig.4 showed the same data of Fig.3 with the statistics added, please merge the two.
- please revise the conclusions.
Major issue:
- Fig. 1A "Western blotting of representative sera samples with HepG2 cell line" there is no HepG2 lane.
- Why the authors titled the autoantibody with an indirect fluorescence? there are methods more reliable and quantitative. Moreover, the IF was presenting only one "positive" and "negative" serum, what about the other seven tested?
- How will performed the ROC analysis between HCC and the other liver diseases? are these marker able to discriminate HCC patients as well?
- Crucial, the OD values of the ELISA assays are very low, please disclaim it in the text and try to provide additional proof of their effective presence in the HCC serum of patients.
Best wishes,
the reviewer.
Round 2
Reviewer 2 Report
Dear Authors,
thanks for having implemented this reviewer's comments in your manuscript. The draft is now suitable for publication.
BW